# Impact of Hydrostatic Pressure on Molecular Structure and Dynamics of the Sodium and Chloride Ions in Portlandite Nanopores

**DOI:** 10.3390/ma17092151

**Published:** 2024-05-04

**Authors:** Run Zhang, Hongping Zhang, Meng Chen, Laibao Liu, Hongbin Tan, Youhong Tang

**Affiliations:** 1School of Materials and Chemistry, Southwest University of Science and Technology, Mianyang 621010, China; izhangrun@163.com (R.Z.); liulaibao@swust.edu.cn (L.L.); clcthb@163.com (H.T.); 2School of Mechanical Engineering, Institute for Advanced Study, Chengdu University, Chengdu 610100, China; 3CAS Key Laboratory of Mineralogy and Metallogeny/Guangdong Provincial Key Laboratory of Mineral Physics and Materials, Guangzhou 510640, China; chenmeng@gig.ac.cn; 4Institute for NanoScale Science and Technology, College of Science and Engineering, Flinders University, Adelaide 5042, Australia

**Keywords:** deep sea, portlandite, diffusion, molecular dynamics simulation

## Abstract

In order to address the issues of energy depletion, more resources are being searched for in the deep sea. Therefore, research into how the deep-sea environment affects cement-based materials for underwater infrastructure is required. This paper examines the impact of ocean depth (0, 500, 1000, and 1500 m) on the ion interaction processes in concrete nanopores using molecular dynamics simulations. At the portlandite interface, the local structural and kinetic characteristics of ions and water molecules are examined. The findings show that the portlandite surface hydrophilicity is unaffected by increasing depth. The density profile and coordination number of ions alter as depth increases, and the diffusion speed noticeably decreases. The main cause of the ions’ reduced diffusion velocity is expected to be the low temperature. This work offers a thorough understanding of the cement hydration products’ microstructure in deep sea, which may help explain why cement-based underwater infrastructure deteriorates over time.

## 1. Introduction

Deep sea exploration has attracted great attention as an effective strategy to discover more resources as global resources are being increasingly depleted [1,2,3]. In recent times, several energy production facilities, including tidal power conversion stations and offshore wind power installations, are venturing beyond shallow water areas into the depths of the ocean. However, constructing resilient infrastructure capable of withstanding the challenging environmental conditions of the deep sea is inherently intertwined with the exploration of these oceanic depths [4,5,6]. Generally, “deep sea” refers to the marine environment with an ocean depth below 200 m. Although cement concretes have been widely used in various marine engineering due to their superior durability and economic efficiency [7,8,9], their service performance and durability under a deep-sea environment has seldom been reported. Currently, most researchers focus on the interactions between seawater ions and concrete in shallow sea areas [10,11]. Due to the variation in sea depth, the physiochemical properties of seawater ions, together with the cementitious materials, change to a great extent. Previous research indicates that ions present in seawater, including calcium, carbonate, magnesium, sulfate, sodium, and chloride, penetrate hardened concrete, leading to significant structural weakening. This phenomenon compromises the durability of concrete structures, particularly in shallow sea environments [12,13,14]. Furthermore, ions are not the only elements affecting the longevity of cement-based materials in deep-sea environments; high hydraulic pressure, low temperature, and somewhat lower pH are all significant influences [15,16,17].

Investigating and forming conclusions about the behavior of cementitious materials, or concrete, after extended exposure to deepwater environments is crucial for selecting the right building materials for deepwater infrastructure [18,19]. Relatively few experimental studies have investigated the influence of ocean depth variations on hydration products. In one notable study, eighteen concrete spheres were submerged at depths ranging from 550 to 1500 m for a period of ten years, commencing with a research initiative by the U.S. Naval Civil Engineering Laboratory in California in 1971. The results revealed that specimens positioned below 1100 m were crushed, likely due to the hydraulic pressure encouraging the dissolution of the concrete specimens [20]. According to Kobayashi et al., Portland cement mortars exposed to a seafloor environment at a depth of 1680 m for 608 days exhibited significant degradation. It was believed that one significant factor contributing to the disintegration of the mortar specimens was the extremely low temperature [21]. Kawabata et al. found that hydraulic pressure has a size effect, with larger specimens experiencing more severe damage when subjected to short-term high hydraulic pressure [22]. Wang et al. reported that water penetration during pressurization may cause microstructural damage to concrete [23]. Following 309 days of exposure at a depth of 3515 m, a specimen’s compressive strength decreased by 27.7%, as assessed by Takahashi et al. [24]. These investigations primarily focused on the impact of low temperature and hydraulic pressure on the mechanical characteristics and microstructure of cement-based materials. However, the influence of ocean depth on the interaction mechanism between ions and cement cannot be adequately explained by these costly, time-consuming, and limited macroscopic experiments. The degradation process of cement-based goods in shallow marine environments has become almost evident due to studies undertaken in recent years [25,26,27]. Hazardous ions gradually infiltrate cement-based materials when exposed to water, leading to the deterioration of concrete structure durability. Through the utilization of the ^35^Cl NMR relaxation approach, Yu et al. examined the adsorption of chloride ions on the surface of cement hydrate phases and observed rapid exchange with free chloride in the bulk solution. Additionally, monosulfide sulfoaluminate (AFm) and portlandite (Ca(OH)_2_) exhibited relatively strong chloride-binding capacities [28]. Employing molecular dynamics modeling, Hou et al. explored the adsorption characteristics of NaCl solution in portlandite nanopores, noting that the hydroxyl group of the portlandite surface enhanced sodium ion adsorption, which was attributed to the formation of surface ionic clusters [29]. Zhou et al. investigated the transport and adsorption characteristics of chloride ions on C-S-H nanopores, discovering that C-S-H with high Ca/Si ratios could bind more chloride ions [30]. Tu et al. examined the adsorption behavior of common ions near a realistic C-S-H gel surface and determined that coupled ion types in solution influenced the ion’s adsorption strength onto C-S-H gels [31].

In light of the preceding discussion, there is a paucity of studies examining how ocean depth influences the adsorption of ions in the various nanopores of cement hydration products. In this work, a portlandite (Ca(OH)_2_) nanopore was constructed to study the adsorption properties of ions in the nanopore under four different ocean depths via molecular dynamics (MD) simulation. A plausible nanoscale explanation for macroscopic experiment can be found in how varying ocean depths affected ions’ behaviors. From a nanoscale standpoint, this discovery offers a scientific foundation for the creation of concrete materials with improved endurance in deep-sea environments.

## 2. Methodology

### 2.1. Model Construction

The parameters of the primitive cell were *a* = *b* = 3.59 Å, *c* = 4.91 Å, *α* = *β* = 90°, *γ* = 120° [32]. The oxygen atoms coordinated with the calcium atoms in an octahedral arrangement. No hydrogen bonds were present within the layer. Cleaving occurred along the (001) surface, exposing hydroxyl groups on the surface and forming a calcium hydroxide surface. By modifying the upper and lower substrates, a 5.7 nm wide channel was created after the supercell’s formation [33,34]. The model parameters were *a* = 43.10 Å, *b* = 43.55 Å, *c* = 80.19 Å, and *α = β = γ* = 90°. The 5.7 nm channel was randomly filled with 40 Na^+^ and 40 Cl^−^ ions, as illustrated in Figure 1 after the introduction of a 0.6 mol/L NaCl solution into the nanopore. Initially, each ion was positioned approximately 20 Å away from the substrate to minimize the substrate’s influence on the ions. The adsorption characteristics of the ions were subsequently investigated by analyzing their dynamic and structural behaviors near the surface using MD simulations.

### 2.2. Force Fields and MD Simulation Procedures

Utilizing the ClayFF force fields established by Cygan et al., which have been widely adopted in atomistic interaction research concerning cementitious materials and their derivatives, the MD simulations in this study were performed [35,36]. The ClayFF force field encompasses four primary components governing the interactions between different atoms: bond length expansion potential energy, Van der Waals force, Coulomb force, and bond angle distortion potential energy, as shown in Equations (1)–(5) [37].
(1)$$E_{total}=E_{coul}+E_{VDW}+E_{bonds-trecth}+E_{angle-bend}$$
(2)$$E_{coul}=\frac{e^{2}}{4\pi_{0}}\frac{q_{1}q_{2}}{r^{2}}$$
(3)$$E_{VDW}=D_{0,ij}\left[{\left(\frac{R_{0,ij}}{r_{ij}}\right)}^{12}-2{\left(\frac{R_{0,ij}}{r_{ij}}\right)}^{6}\right]$$
(4)$$E_{bond-strech}=K_{1}{\left(r_{ij}-r_{0}\right)}^{2}$$
(5)$$E_{angle-bend}=K_{2}{\left(\theta_{ijk}-\theta_{0}\right)}^{2}$$

All molecular dynamics simulations were conducted using the GROMACS 2018.8 package, an open-source program [38]. To optimize the energy of the simulated system, the model underwent an initial relaxation period in the simulation. Subsequently, the model was subjected to canonical ensemble (NPT) conditions for 3 ns. In a real deep-sea environment, the hydraulic pressure increases at a rate of 1 MPa for every 100 m of depth [21,39]. The temperature of the sea gradually decreases with increasing ocean depth, dropping rapidly to below 2–3 °C beyond 1000 m [40,41]. This study considered four different water depths: 0, 500, 1000, and 1500 m. Table 1 presents the temperature and specific pressure at each depth. To ensure the simulation’s accuracy, the time step was set to 1 fs. Temperature was controlled using the Nosé–Hoover thermostat, while pressure was regulated using the Parrinello–Rahman Barostat. Equilibrium trajectories of every atom in the model were outputted every 1 ps. Finally, an analysis of the equilibrium dynamic trajectory of each atom was conducted to investigate the interaction of portlandite with ions and water.

## 3. Results

### 3.1. Molecular Structure of Interfacial Water Molecules

The pore size of the portlandite nanopore increased by roughly 3 Å after simulation for 8 ns. The atomic density profiles depict the distribution of water molecules throughout the nanoscale channel perpendicular to the substrate, allowing the interaction between water molecules and the portlandite interface to be investigated. As shown in Figure 2, the density profiles of O_w_ and H_w_ in the nanopore were slightly asymmetric near the interface, which agrees in general with previous works [29]. The slight asymmetry might have originated from the sample size, and a large sample could alleviate the impact on the density profile from localized ions. The density profile for water molecules was only discussed at a depth of 0 m, since the influence of the portlandite nanopore surface on the density profile of water molecules was similar at four different depths. For the density profile of O_w_, it can be observed in Figure 2a that there were four obvious peaks at 14.2, 17.5, 67.5, and 70.8 Å, and the peak value was the highest at 14.2 Å. Similarly, the density profile of H_w_ shows four obvious peaks near the interface. Comparing it with the O_w_, it can be found that the peaks of H_w_ were lower and closer to the interface, indicating that the H atoms of water molecules at the portlandite interface pointed toward the surface. Furthermore, the multiple peaks of density distribution indicate that water molecules were layered near the interface due to the effect of hydroxyl groups on the surface. The maximum peaks of the O_w_ density profile at four depths were 1.15, 1.17, 1.18, and 1.19 g/cm^3^, respectively, demonstrating that increasing depth may improve hydrophilicity in the portlandite interface. The densities of O_w_ and H_w_ in the central region of the channel were 0.83 g/cm^3^ and 0.11 g/cm^3^, respectively. The density of water molecules in this region (0.94 g/cm^3^) was lower than their original density (1.0 g/cm^3^).

The radial distribution function (RDF) was used to investigate the influence of the portlandite surface on water molecules along the z axis. As shown in Figure 3a, the RDF for pairs of O_w_ exhibited a clear peak at 2.78 Å and two weaker peaks at 4.66 Å and 6.76 Å. These three peaks correlated to three hydration shells. The weaker peak was observed because water molecules were influenced by the ions in the solution, which explains the layered accumulation of water molecules at the interface. To study the local structure of water molecules, the RDF of O atoms of water molecules (O_w_) and H atoms of hydroxyl groups (Ho) was also analyzed. Figure 3b shows that the first peak in the RDF of O_w_-Ho occurred at around 1.84 Å, indicating a strong, close-range spatial correlation for the two atoms. Simultaneously, the positions of the first valley value of O_w_-O_w_ and the second valley of O_w_-Ho gradually moved to the left as the depth increased, indicating that the local structure of the water molecules was perturbed by the depth change to some extent.

### 3.2. The Local Structure of Chloride and Sodium Ions

By analyzing the density profiles of ions in the portlandite channel, the distribution and adsorption behavior of ions at four depths could be explored. As shown in Figure 4, the density curve of Cl ions exhibited a prominent, high-intensity peak, but the curve of Na ions was relatively weak closer to the interface. Previous studies have shown that sodium atoms remain on the substrate for a longer time [29,42]. The vibration of the surface hydroxyl groups caused local negative and positive charge elements to develop on the portlandite surface, resulting in the adsorption of Na and Cl ions [43,44]. Additionally, the density intensity of the bottom surface was greater than that of the upper surface, indicating that the bottom surface had a large quantity of ion accumulation. Furthermore, the difference in density between the two surfaces showed the random selection of ions, excluding the influence of the substrate on ions. The curve of the Cl ions was shifted away from the interface by about 2 Å compared to the Na ions, indicating that Na ions were directly adsorbed to the surface, whereas Cl ions were adsorbed indirectly. This result agrees with that reported by Hou et al. [29]. The surface of portlandite may absorb sodium ions due to the hydroxyl group’s negative charge. Interestingly, the depth change affects the magnitude of the ionic density intensity peak, but not its placement. Figure 4 shows that the ion distribution at the center of the channel was identical. In Figure 4a,b, the peak value of the Na ions was greater than that of the Cl ions. On the contrary, the peak value of Cl ions was greater than that of the Na ions in Figure 4c,d.

As shown in Table 2, the number of ions within the 5 Å range from the interface at four depths during the simulation period was counted. The adsorption rate of Na ions and Cl ions reached 23.78% and 39.13% at a depth of 1500 m, respectively, the highest rate compared to the other three depths. At the same time, the adsorption rate of chloride ions at the interface was significantly higher than that of sodium ions at all four depths, indicating that the surface has a large adsorption capacity for chloride ions. The portlandite has significant chloride ion adsorption capabilities, which is primarily due to two factors. First, the hydroxyl group on the surface of calcium hydroxide can form a hydrogen bond with chloride ions. Second, more sodium ions are stably adsorbed on the surface, and sodium ions can be combined with chloride ions to increase the surface packing density of chloride ions.

Figure 5 shows the diffusion statuses of ions at different depths. In contrast to the initial model in Figure 1, both Na and Cl ions accumulated at the predicted interface. At the same time, it is obvious that the Na ions were closer to the interface than the Cl ions, indicating that the Na ions had a higher adsorption capability on the surface of portlandite (yellow lines). Meanwhile, when the ocean depth increased from 0 to 1000 m, temperature and hydraulic pressure exerted a coupling impact on the ions’ local structural variations. The analysis of the density profiles, ion interface adsorption rates, and snapshots during this process indicate that increasing the pressure can improve the ion adsorption capacity of the portlandite surface. Additionally, Ca ions and hydroxyl groups can be identified in the pore channel, indicating that portlandite will partially dissolve when interacting with ions (Figure 5).

As shown in Figure 6, the representative snapshots of the portlandite model at 8 ns indicate that ions in solution formed clusters with ions. When hydrogen atoms (Ho) of the nearest four hydroxyl groups were scattered, the oxygen atom (O_h_) was exposed to form a negative charge, which adsorbed Na ions, as shown in Figure 6a. On the other hand, as illustrated in Figure 6b, if the Ho of the four hydroxyl groups was clustered, a local positive charge was formed, which adsorbed Cl ions. Because of the small radius, Na ions quickly traversed the space between the hydroxyl groups and formed Na-O_h_ bonds. The persistent bond between surface hydroxyl groups and Na ions permitted Na ions to exist for an extended period of time. This phenomenon is reflected in the density profiles, with Na ions generating an evident peak at the interface. In contrast to Na ions, Cl ions with greater atomic radii could only diffuse near the surface. The adsorbed Cl ion was bonded to the hydroxyl group on one side and surrounded by water molecules on the other. The adsorption states of different ions on the surfaces can cause considerable differences in their dynamical properties.

The radial distribution function is used to explain the local structure of the ions near the interface and in the pore solution in great detail. According to Figure 7a, there are two pronounced peaks at 2.36 Å and 4.54 Å for the RDF of Na-O_w_. The first peak at 2.36 Å, corresponding to the bond length of Na-O_w_, is in a good agreement with the experimental data (2.3–2.5 Å). The first peak of Cl-O_w_ appears at 3.24 Å and corresponds to the length of the Cl-O_w_ bond. As such, there are more Na ions surrounding water molecules than Cl ions. Simultaneously, there are more weak peaks in the RDF of Cl-O_w_, demonstrating that the spatial ordering of Cl ions with water molecules can be extended to larger distances in Figure 7b. It is observed that depth does not affect the spatial correlation between water molecules and ions (Na and Cl).

In Figure 7c, the first peak of RDF arises near 2.36 Å, corresponding to the bond length of the Na-O_h_ pair. It is the first peak that is relatively sharp, suggesting that Na ions can form stable bonds with structural O atoms. This result indicates that Na ions can be stably adsorbed on the portlandite surface. The Na-O_h_ profile still has many weak peaks in a long range, indicating a strong spatial correlation of Na ions with O_h_. This observation explains the firm adsorption of sodium ions on the surface. The height of the Na-O_h_ peak gradually diminishes as the depth increases from 0 to 1000 m, which is saturated when the depth further increases to 1500 m. This means that the spatial restriction of Na ions with O_h_ steadily grows with depth and does not change once it reaches a specific depth. Figure 7d depicts the RDF curve of Cl-O_h_, which exhibits an obvious peak around 3.24 Å. Compared with the Na-Oh RDF curve, the peak for Cl-O_h_ shifts slightly to the right, indicating that the link between Na and O_h_ is shorter and more stable. It should be noticed that the first peak of Na-O_h_ in the RDF curve is higher than that of Cl-O_h_, implying that there are more Na ions surrounding the hydroxyl groups.

The RDF curve of Na-Cl has two distinct peaks, as shown in Figure 7e. The first peak appears at 2.80 Å, corresponding to the Na-Cl bond. The second peak appears at 5.02 Å, and is regarded as the Na-O_w_-Cl pair. After forming a hydration layer with the surrounding water molecules, Na attracts Cl to establish coordination. The first peak of the RDF of Na-Cl becomes much lower at 1500 m, whereas the second peak grows significantly higher, forecasting that the connection pattern of Na and Cl ions at this depth tends to be the Na-O_w_-Cl pair. The first RDF peak of Cl-Ho appears at 2.34 Å in Figure 7f. When compared to Cl-O_h_, the Cl-Ho peak shifts to the left, indicating that Cl ions are more likely to form bonds with Ho of hydroxyl groups, as seen by the spatial structure in Figure 6b. And the peak of Cl-Ho is higher than that of Cl-O_h_, which could be explained by the fact that the coordination number of Ho ions for Cl ions is higher than that of O_h_ ions. In general, the position and number of RDF peaks stays nearly constant at different depths, signifying that depth has minor effect on the bond length and spatial structure of each ion pair in the solution. Moreover, it is found that depth has a considerable effect on the peak value of RDF.

The coordination number of the ions reveals the structural arrangement of the ions that determine the interactions between the ions and the interface. Water molecules, O_h_ in hydroxyl groups, and Cl ions in solution are the neighboring atoms of Na ions. The coordination number of Na ions is listed in Table 3. Similarly, water molecules, Ca ions, Ho in hydroxyl groups, and Na ions in solution are the neighboring atoms of Cl ions. The coordination numbers of Cl ions are listed in Table 4. The coordination numbers of Na and Cl ions are in agreement with the previous calculation and experimental results [29,45]. Na-O_h_ has a higher coordination number than that of Cl-Ho, suggesting that Na ions are more easily adsorbed on the portlandite surface than Cl ions. For Na-O_h_ and Cl-Ho bonds, the coordination numbers of ion pairs experience minor changes when the depth varies, implying that the depth variation exerts a minimal effect on the local structure of ions. It is worth mentioning that Cl ions may be accompanied by more water molecules than Na ions, and thus, their total coordination number is slightly higher. Such an observation is reasonable, as Cl ions’ hydration radius is larger than that of Na ions.

### 3.3. Dynamic Properties of Water Molecules and Ions

The dynamic properties of various chemical bonds are described by the time correlation function (TCF) [36]. The stability of chemical bonds can be assessed by comparing the TCF curves of different ion pairs. The equation is as follows:(6)$$C(t)=\frac{\langle \delta b(t)\delta b(0)\rangle }{\langle \delta b(0)\delta b(0)\rangle }$$
where δb(t)=b(t)-〈b〉, *b(t)* is a binary operator. At time *t*, if the distance of ions–ions or ions–water molecules is within the cut-off radius (the position of the first valley of RDF), the TCF value is 1; otherwise, it is 0. The closer the TCF value is to 1, the more durable the bond is. The chemical bond stability for different pairs is listed in the following order: Na-O_h_ > Na-Cl > Na-O_w_ > Cl-O_w_. Figure 8a,b show that the stability levels of Na-O_w_ and Cl-O_w_ decrease to 0.3 after 1000 ps, showing that their stability is extremely weak. At the same time, the stability curves of Na-O_w_ and Cl-O_w_ are extremely similar at all four depths, indicating that depth has minimal impact on their stability. Figure 8c shows that the stability of Na-O_h_ is significantly higher than that of Na-O_w_, confirming that there is a strong interaction between Na ions and the portlandite surface, which is also why Na ions can stably adsorb on the portlandite surface. At the same time, the stability of Na-O_h_ decreases with increasing depth. On the contrary, the stability of Na-Cl shows an increasing trend with increasing depth in Figure 8d. Overall, different ion pairs respond differently to depth variations.

The mean square displacement (*MSD*) can be used to calculate the speed of movement of water molecules or ions in a system, which is useful for analyzing its kinetic properties [46]. The formula is as follows:(7)$$MSD(t)=\sum_{i-1}^{n}\langle {\left|r_{n}(t)-r_{n}(0)\right|}^{2}\rangle$$
where *r_n_* (*t*) is the position of atom n at time t, and *r_n_* (0) is the initial position of atom n. A higher *MSD* value suggests more violent motion and faster diffusion for various atoms. In addition, the diffusion coefficient *D* is calculated by linearly fitting the *MSD* [47]. The diffusion coefficient quantifies the dynamic properties of water molecules and ions in the channel. The formula is as follows:(8)$$D=\frac{MSD}{6t}$$

The *MSD* curves of the ions increase linearly and almost coincide during the 300 ps, demonstrating that the starting state of motion of the distinct ions in the solution stays randomly and is unaffected by the substrate. However, after 300 ps, the ions display distinct motion characteristics. Figure 9 depicts how the *MSD* of water and ions varies with increasing depth. It can be found the *MSD* relationship of water > Cl^−^ > Na^+^. Water molecules exhibit a larger *MSD* than that of Na and Cl ions in solution.

Water molecules are ion carriers; hence, their transport rate is often greater than that of ions. Furthermore, the transfer rate is lower than that of Cl ions because Na ions are more easily adsorbed by the O_h_ of the hydroxyl group. The MSD of water molecules, Na ions, and Cl ions decreases somewhat with increasing depth. The MSD curves at 1000 m and 1500 m depths are found to be close to each other throughout the simulation. When combined with the preceding analysis, the MSD curves of water molecules and ions at the two depths practically overlap, owing to the close proximity of the temperatures at the two depths. Overall, pressure has little effect on ion diffusion, and low temperature is the primary factor reducing the ion diffusion rate. The MSD of hydroxyl groups in the substrate is substantially smaller than that of water molecules and ions in solution, indicating that hydroxyl groups in the substrate are constrained, as shown in Figure 9d. It has been discovered that increasing pressure and decreasing temperature can accelerate the diffusion rate of hydroxyl groups. However, when the temperature remains constant and the pressure continues to rise, the diffusion rate of hydroxyl groups is inhibited.

The diffusion coefficients of water molecules and ions in the portlandite nanopore are shown in Table 5. The diffusion coefficients decline in order of water > Cl^−^ > Na^+^ > OH^−^ > Ca^2+^. With the depth increasing from 0 m to 1500 m, the diffusion coefficient of Na ions decreases remarkably from 1.26 to 0.69 × 10^−9^ m^2^·s^−1^, which is the most significant. Added to that, when compared to Ca ions, the diffusion coefficient of OH group that is exposed on the surface is larger. In short, higher pressure and lower temperature slow the transport of water molecules and ions. The decrease in temperature accelerates the disintegration rate of the portlandite substrate, which agrees with the results reported by Kobayashi et al. [21].

To further study the influence of variations in depth on the dynamic properties of Na and Cl ions, the minimum distance from ions to hydroxyl groups and the number of interactions within a particular distance between ions and hydroxyl groups were estimated. As shown in Figure 10, Cl ions reach the area of hydroxyl groups in less time than Na ions, demonstrating that Cl ions travel more quickly. Figure 10a shows that the duration for Na ions to reach the hydroxyl groups increases from 0 to 1000 m and decreases to some extent from 1000 to 1500 m. In contrast, Figure 10b demonstrates that the time required for Cl ions to reach the hydroxyl groups reduces from 0 to 1000 m and then increases from 1000 to 1500 m. These observations suggest that temperature reduction is the major cause of the change in the time required for Na ions and Cl ions to come to the hydroxyl groups.

As illustrated in Figure 11, the number of interactions between Na ions and hydroxyl groups at a given distance is greater than that between Cl ions and hydroxyl groups, suggesting more Na ions and hydroxyl interactions. In other words, there are more Na ions adsorbed on the surface of portlandite. To summarize, both low temperature and high pressure can alter the dynamic properties of ions, while temperature is the dominant factor.

## 4. Conclusions

In this study, molecular dynamics simulation was used to explore the effect of depth on the local structure and dynamic properties of water molecules and ions at the portlandite interface. The hydrolysis of water molecules at the interface remains unaffected by variations in depth. The density distribution of water molecules within the nanopore remains constant, indicating that depth variation has only a minor impact on the local structure of these molecules. It should be noted that the density profile and interfacial adsorption rate of ions in the pore channel do exhibit variations with changes in depth, but the change is small. Increasing depth has an ignorable effect on the bond length or spatial structure of ion pairs, but it does modify the ion’s coordination number. Meanwhile, as the depth increases, the stability of Na-O_w_, Cl-O_w_, and Na-Cl bonds increases, while the stability of Na-O_h_ bonds decreases. With increasing depth, the diffusion rates of water molecules, Na ions, and Cl ions reduce. The temperature is identical, while the pressure varies between 1000 and 1500 m. The diffusion rates of molecules and ions at 1000 and 1500 m depths are similar, implying that pressure variations do not influence ion motion properties. Further investigation reveals that high pressure and low temperature may considerably alter the migratory properties of water and ions, with temperature playing a primary role.

## Figures and Tables

**Figure 1 materials-17-02151-f001:**
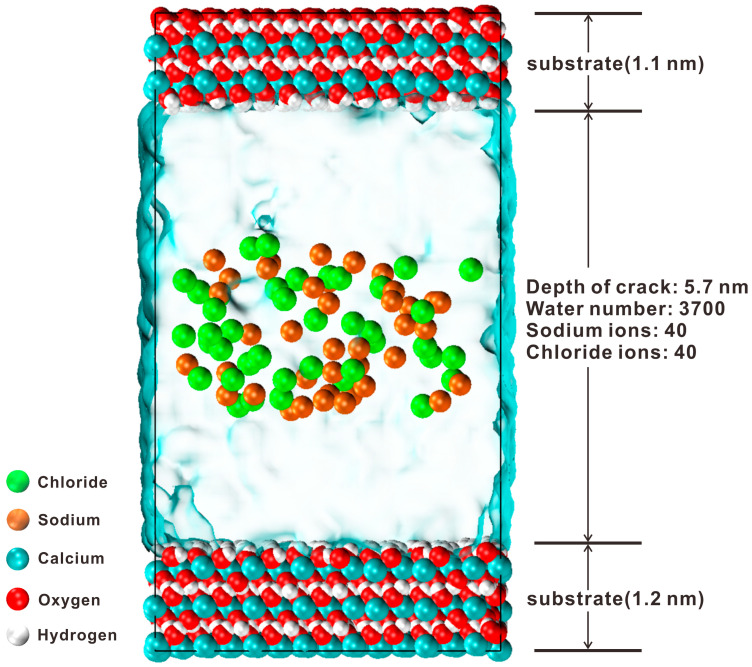
The model of portlandite nanopore in which differently colored spheres represent different atoms or ions.

**Figure 2 materials-17-02151-f002:**
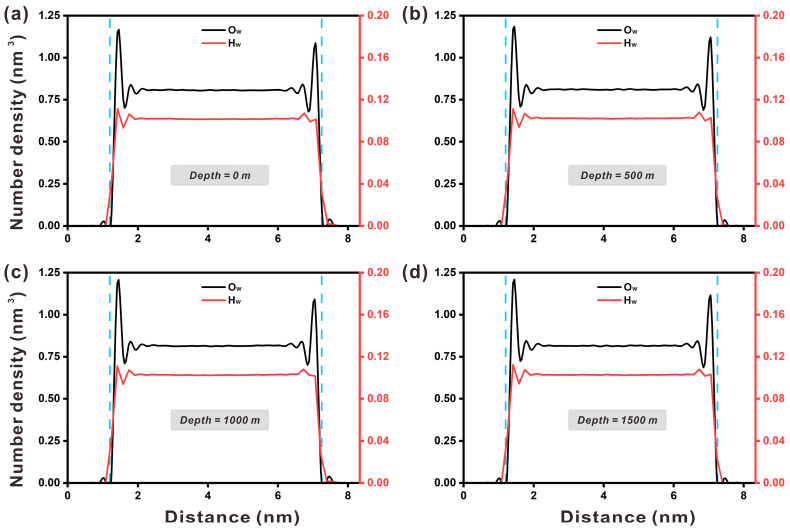
Density profiles for water oxygen (O_w_) and water hydrogen (H_w_) at four depths: (**a**) depth at 0 m, (**b**) depth at 500 m, (**c**) depth at 1000 m, and (**d**) depth at 1500 m (blue lines represent surface lines).

**Figure 3 materials-17-02151-f003:**
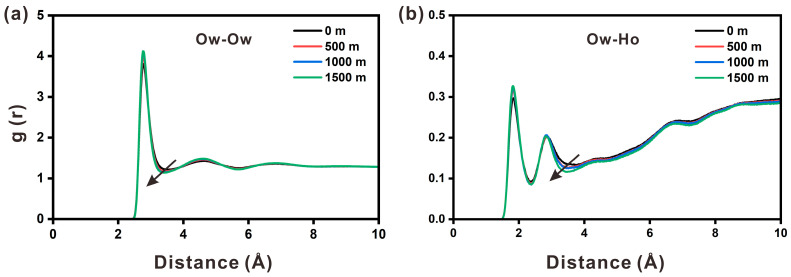
The radial distribution function between oxygen atoms (O_w_) of water molecules and oxygen atoms (O_w_) of water molecules (**a**), or hydrogen atoms (Ho) of hydroxyl group (**b**), in the portlandite nanopore at four different depths.

**Figure 4 materials-17-02151-f004:**
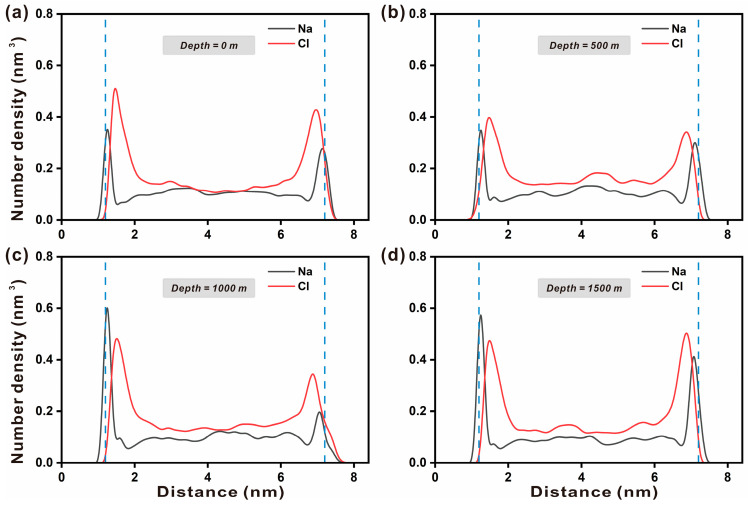
Density profiles for Na and Cl ions at fourdepths: (**a**) depth at 0 m, (**b**) depth at 500 m, (**c**) depth at 1000 m, and (**d**) depth at 1500 m (blue lines represent surfaces).

**Figure 5 materials-17-02151-f005:**
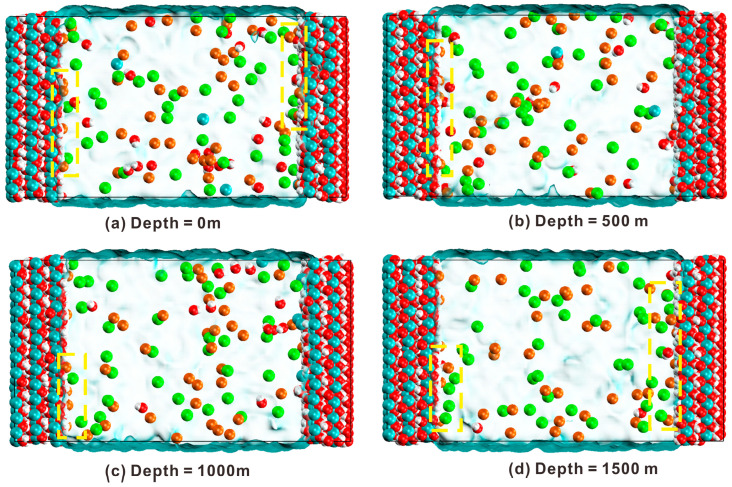
Snapshots of the adsorption of ion transport in the nanopores of portlandite at 8000 ps at four depths: (**a**) depth at 0 m, (**b**) depth at 500 m, (**c**) depth at 1000 m, and (**d**) depth at 1500 m (yellow wire frame indicates the local atomic structure).

**Figure 6 materials-17-02151-f006:**
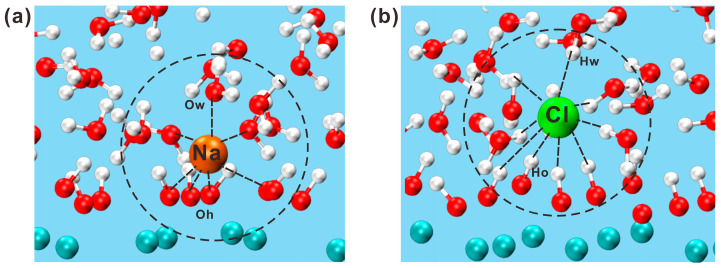
The schematic diagram of interaction between ions and interface in the solution: (**a**) Na ions and (**b**) Cl ions (black circle indicates the bonding of atoms).

**Figure 7 materials-17-02151-f007:**
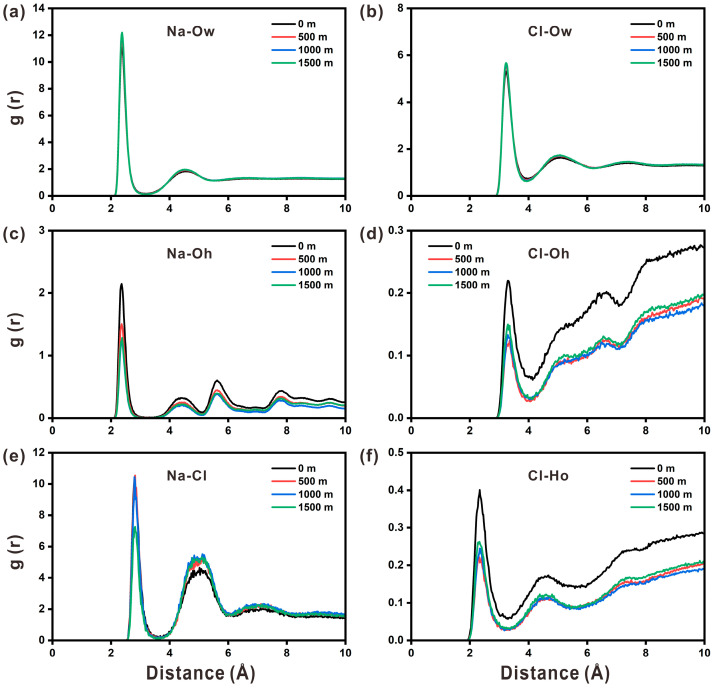
The radial distribution function for (**a**) Na-O_w_, (**b**) Cl-O_w_, (**c**) Na-O_h_, (**d**) Cl-O_h_, (**e**) Na-Cl, and (**f**) Cl-Ho at four different depths.

**Figure 8 materials-17-02151-f008:**
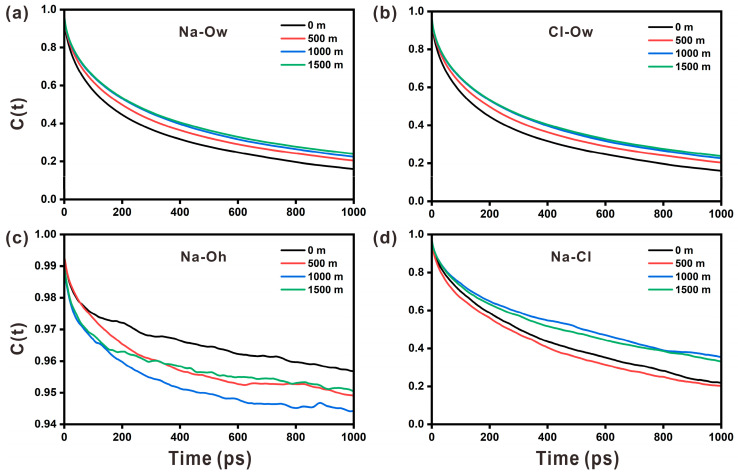
The time correlation function for (**a**) Na-Ow, (**b**) Cl-Ow, (**c**) Na-Oh, and (**d**) Na-Cl pairs at different depths.

**Figure 9 materials-17-02151-f009:**
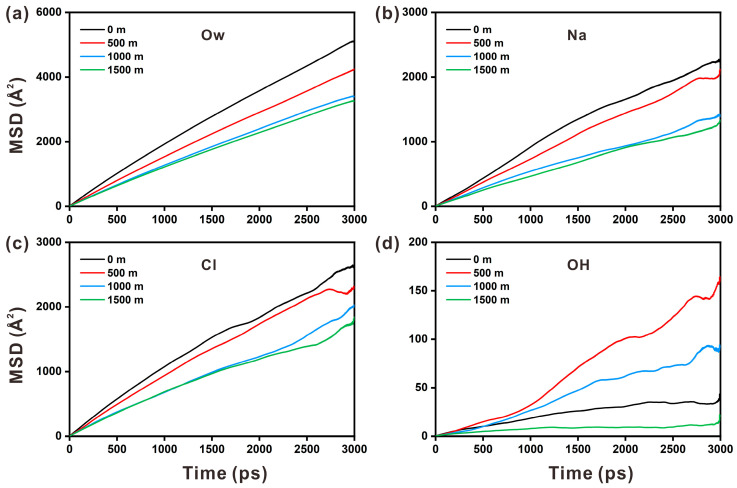
The mean square displacement variations for (**a**) water molecules (O_w_), (**b**) Na ions, (**c**) Cl ions, and (**d**) hydroxyl groups (OH) at different depths.

**Figure 10 materials-17-02151-f010:**
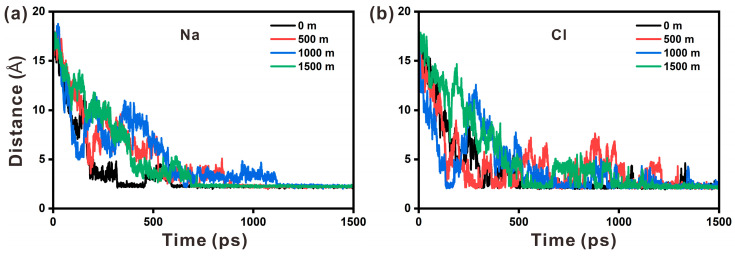
The minimum distance changes between (**a**) Na ions and hydroxyl groups and (**b**) Cl ions and hydroxyl groups at different depths.

**Figure 11 materials-17-02151-f011:**
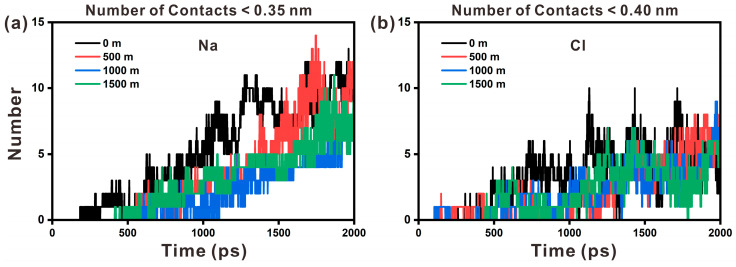
The number of contacts between (**a**) Na ions and hydroxyl groups and (**b**) Cl ions and hydroxyl groups at different depth.

**Table 1 materials-17-02151-t001:** The pressure and temperature of four ocean depths.

Depth (m)	Pressure (MPa)	Temperature (K)
0	0.1	297
500	5	283
1000	10	277
1500	15	276

**Table 2 materials-17-02151-t002:** Ion adsorption rates within 5 Å range from the interface at four depths.

Ions	0 m	500 m	1000 m	1500 m
Na^+^	20.22%	20.77%	16.42%	14.82%
Cl^−^	38.25%	30.10%	35.54%	13.15%

**Table 3 materials-17-02151-t003:** The coordination numbers of Na ions at four depths.

Depth (m)	Cl	O_w_	O_h_	Total
0	0.02	5.24	0.44	5.70
500	0.02	5.20	0.30	5.52
1000	0.02	5.27	0.23	5.52
1500	0.01	5.28	0.24	5.53

**Table 4 materials-17-02151-t004:** The coordination numbers of Cl ions at four depths.

Depth (m)	Na	O_w_	Ho	Ca	Total
0	0.02	7.33	0.18	0.11	7.64
500	0.02	7.33	0.10	0.11	7.56
1000	0.02	7.28	0.10	0.12	7.52
1500	0.01	7.33	0.11	0.12	7.57

**Table 5 materials-17-02151-t005:** The diffusion coefficients of ions at four depths (10^−9^ m^2^/s).

Depth (m)	O_w_	Na	Cl	Ca	OH
0	2.77	1.26	1.36	0.008	0.021
500	2.30	1.16	1.35	0.087	0.095
1000	1.90	0.71	0.97	0.050	0.054
1500	1.79	0.69	0.86	0.001	0.004

## Data Availability

Data are contained within the article.

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
