# Peer review of "Impact of Hydrostatic Pressure on Molecular Structure and Dynamics of the Sodium and Chloride Ions in Portlandite Nanopores"

_materials, 2024, doi:10.3390/ma17092151_

Round 1

Reviewer 1 Report

Comments and Suggestions for Authors

See uploaded PDF

Additionally: 

The authors should discuss previous studies regarding fluids in channels, specifically J. Chem. Theory Comput 2012, 8, 2012-2022 and J. Chem Theory Comput. 2011, 7, 1736-1749.  These prior studies used a combination of atomistic and continuum treatments. Specifically, these prior works developed a model for including electric fields in MD using an atomistic-to-continuum framework, which provides the mathematical and the algorithmic infrastructure to couple finite element (FE) representations of continuous data with atomic data. The authors should note that this previous work represents the electric potential on a FE mesh satisfying a Poisson equation with source terms determined by the distribution of the atomic charges. In regard to synergistic areas with the submitted paper, the previous works carried out a calculation of a salt water solution in a silicon nanochannel to demonstrate the method in a target scientific application in which ions are attracted to charged surfaces in the presence of electric fields and interfering media.

Author Response

In this submission to Materials, the authors examined the impact of ocean depth (0, 500, 1000, and 1500 m) on the ion interaction processes in concrete nanopores using molecular dynamics simulations. The authors find that at the portlandite interface, the local structural and kinetic characteristics of ions and water molecules are examined. The authors' findings show that the portlandite surface hydrophilicity is unaffected by increasing depth. The authors find that the density profile and coordination number of ions alter as depth increases, and the diffusion speed noticeably decreases. The authors conclude that their work offers a thorough understanding of the cement hydration products' microstructure in the deep sea, which may help explain why cement-based underwater infrastructure deteriorates over time.

I find this manuscript to be of interest to researchers specializing in fluids in nanopores as well as readers of Materials. As such, I am supportive of publication with a few edits. pecifically, there has been prior work investigating fluids in channels, which should be noted: J. Chem. Theory Comput. 2012, 8, 2012-2022 and J. Chem. 2012,8,2012-2022; Theory Comput. 2011, 7, 1736-1749.2011,7,1736-1749。Specifically, these prior studies examined the effects of fluid (both at the atomistic and continuum fluid level) to investigate these systems. With this minor note, I would be willing to re-review this manuscript for subsequent publication in Materials.

The authors should discuss previous studies regarding fluids in channels, specifically J. Chem. Theory Comput 2012, 8, 2012-2022 and J. Chem Theory Comput. 2011, 7, 1736-1749. These prior studies used a combination of atomistic and continuum treatments. Specifically, these prior works developed a model for including electric fields in MD using an atomistic-to-continuum framework, which provides the mathematical and the algorithmic infrastructure to couple finite element (FE) representations of continuous data with atomic data. The authors should note that this previous work represents the electric potential on a FE mesh satisfying a Poisson equation with source terms determined by the distribution of the atomic charges. In regard to synergistic areas with the submitted paper, the previous works carried out a calculation of a salt water solution in a silicon nanochannel to demonstrate the method in a target scientific application in which ions are attracted to charged surfaces in the presence of electric fields and interfering media.

Reply: Thanks for your comments.  These helpful works  have been cited in the revised manuscript. Besides, we added the following contents in the introduction section of the revised manuscript:

“The vibration of the surface hydroxyl groups causes local negative and positive charge elements to develop on the portlandite surface, resulting in the adsorption of Na and Cl ions. [43, 44]

Reviewer 2 Report

Comments and Suggestions for Authors

Two short comments to the Authors:

The numbering of references should be revised since Ref[2] (line57) does not match the list.

Line 208 - How do you think, the diameter of Na and Cl ions play a role in your calculations, doesn't it?

Author Response

  1. The numbering of references should be revised since Ref[2] (line57) does not match the list.

Reply: We would like to thank the reviewer for pointing this out. We have carefully revised the manuscript and the corresponding changes have been made in the updated manuscript.

  1. Line 208 - How do you think, the diameter of Na and Cl ions play a role in your calculations, doesn't it?

Reply: Thanks for your comment. Because the radius of sodium ions is smaller than that of chloride ions, sodium ions are more likely to approach the CH/water interface and form stable coordination relationships with surface hydroxyl groups. This is also why the intensity peak of the sodium ion density distribution curve mainly appears at the interface.

Reviewer 3 Report

Comments and Suggestions for Authors

The authors discuss the influence of hydrostatic pressure on the molecular structure in portlandite nanopores. Based on studies of the influence of ocean depth and pressure there, they found that increasing the depth had no effect on the hydrophilicity of the portlandite surface. They conclude that the main cause of the reduced ion diffusion rate will be low temperature.

The work is generally very well designed and has high scientific potential. I support the implementation of the project.

Author Response

The authors discuss the influence of hydrostatic pressure on the molecular structure in portlandite nanopores. Based on studies of the influence of ocean depth and pressure there, they found that increasing the depth had no effect on the hydrophilicity of the portlandite surface. They conclude that the main cause of the reduced ion diffusion rate will be low temperature.

The work is generally very well designed and has high scientific potential. I support the implementation of the project.

Reply: We greatly value your insightful feedback and recommendations. As a result, significant enhancements have been made to the language and readability of the manuscript. We sincerely trust that this revised version achieves a heightened linguistic standard and enhances the clarity of our argument.